# RHO GTPase-Related Long Noncoding RNAs in Human Cancers

**DOI:** 10.3390/cancers13215386

**Published:** 2021-10-27

**Authors:** Mahsa Saliani, Amin Mirzaiebadizi, Niloufar Mosaddeghzadeh, Mohammad Reza Ahmadian

**Affiliations:** 1Institute of Biochemistry and Molecular Biology II, Medical Faculty and University Hospital Düsseldorf, Heinrich-Heine University, 40225 Düsseldorf, Germany; mahsa.saliani@hhu.de (M.S.); aminmirzaie1992@gmail.com (A.M.); mosaddeg@uni-duesseldorf.de (N.M.); 2Department of Chemistry, Faculty of Science, Ferdowsi University of Mashhad, Mashhad 9177948974, Iran

**Keywords:** carcinogenesis, lncRNAs, miRNAs, RHO GTPases, signal transduction, sponge effect

## Abstract

**Simple Summary:**

Whilst mutations in genes encoding RHO GTPase proteins are rare in different type of cancers, altered expression of several RHO GTPases has been reported in a variety of human malignancies. As key regulators of gene expression, lncRNAs coordinate a wide range of molecular processes, including post-translational regulation through miRNA sponging. The purpose of the present study was to address the current state of knowledge about the lncRNAs involved in the regulation of the expression of the RHO GTPases including RHOA, RHOB, RHOC, RAC1, and CDC42, with a specific focus on the regulatory mechanism of lncRNAs as the molecular sponges of miRNAs. Considering the critical roles of lncRNAs in malignancies, lncRNA-based therapeutics are representing promising approaches in cancer treatment through novel technologies. In this regard, well-characterized examples of lncRNAs associated with tumorigenicity present experimental frameworks for future studies in this rapidly evolving field.

**Abstract:**

RHO GTPases are critical signal transducers that regulate cell adhesion, polarity, and migration through multiple signaling pathways. While all these cellular processes are crucial for the maintenance of normal cell homeostasis, disturbances in RHO GTPase-associated signaling pathways contribute to different human diseases, including many malignancies. Several members of the RHO GTPase family are frequently upregulated in human tumors. Abnormal gene regulation confirms the pivotal role of lncRNAs as critical gene regulators, and thus, they could potentially act as oncogenes or tumor suppressors. lncRNAs most likely act as sponges for miRNAs, which are known to be dysregulated in various cancers. In this regard, the significant role of miRNAs targeting RHO GTPases supports the view that the aberrant expression of lncRNAs may reciprocally change the intensity of RHO GTPase-associated signaling pathways. In this review article, we summarize recent advances in lncRNA research, with a specific focus on their sponge effects on RHO GTPase-targeting miRNAs to crucially mediate gene expression in different cancer cell types and tissues. We will focus in particular on five members of the RHO GTPase family, including RHOA, RHOB, RHOC, RAC1, and CDC42, to illustrate the role of lncRNAs in cancer progression. A deeper understanding of the widespread dysregulation of lncRNAs is of fundamental importance for confirmation of their contribution to RHO GTPase-dependent carcinogenesis.

## 1. Introduction

RHO family proteins are GDP/GTP-binding proteins of the RAS superfamily that transduce extracellular signals into intracellular responses. They cycle, with some exceptions, between an inactive GDP-bound (“off”) state and an active GTP-bound (“on”) state (Figure 1) [1,2]. Fully processed GTPases remain locked in an inactive GDP-bound form in the cytosol via the formation of stoichiometric complexes with RHO GDP dissociation inhibitors (GDIs) [3]. During the signal transduction to its downstream effectors to activate multiple signaling pathways, most RHO GTPases acquire their active state with the exchange of GDP for GTP as a consequence of two continuous steps. The first step involves their release from GDI complexes, followed by rapid GDP/GTP exchange, a process catalyzed by RHO-specific guanine nucleotide exchange factors (GEFs) [1,2,4,5,6]. In the active state, conformational changes of two flexible regions, called switch I and switch II, provide a functional platform for the interaction of RHO GTPases with their regulators and effectors [7,8]. At the end of the GTPase cycle, hydrolysis of the bound GTP is stimulated by GTPase-activating proteins (GAPs) and converts RHO GTPases to the GDP-bound inactive state [2,9,10]. RhoGDI can extract both inactive and active RhoGTPases, and found that extraction of active RhoGTPase contributes to their spatial regulation around cell wounds [9].

As with most RAS superfamily proteins, the maturation of the newly translated RHO GTPases requires a stepwise posttranslational modification. This modification includes the incorporation of a geranylgeranyl moiety (in some cases, farnesyl or palmitate groups) onto the CAAX box (C is cysteine, A is any aliphatic amino acid, and X is any amino acid) on the endoplasmic reticulum [10,11], the subsequent cleavage of the AAX tripeptide, and the methylation of the newly exposed C-terminal isoprenylcysteine residue [12]. In stimulated cells, isoprenylated RHO GTPases are specifically associated with the cellular membrane, which is essential for their biological activity (Figure 1). In resting cells, however, GDIs extract them from the membrane and create a cytosolic pool of inactivated RHO GTPases [13].

To date, 20 canonical members of the RHO family have been identified in humans and can be categorized into distinct subfamilies based on their sequence homology [2,14]: RHO (RHOA, RHOB, and RHOC); RAC (RAC1, RAC1B, RAC2, RAC3, and RHOG); CDC42 (CDC42, G25K, TC10/RHOQ, TCL/RHOJ, WRCH1/RHOU, and WRCH2/RHOV); RHOD (RHOD and RIF/RHOF); RND (RND1/RHO6, RND2/RHO7, and RND3/RHO8/RHOE); and TTF/RHOH [15]. Some studies also include the members of the RHOBTB and Miro families among those proteins; however, these ‘atypical’ GTPases are highly divergent from the rest of the RHO GTPases in terms of structure, overall amino acid homology, subcellular localization, and biological functions [16].

Activated RHO GTPases bind to their downstream effectors and thereby regulate diverse cellular processes (Figure 1), including the reorganization of the actin cytoskeleton and thereby cell adhesion, polarity, and migration [17,18]. Thus, they are associated with the control of various biological processes, such as cell trafficking, wound healing, immune response, embryonal development, and neuronal plasticity [19,20]. Given the critical roles of RHO GTPases in cell signaling, the deregulation of their downstream pathways is known to contribute to the development of diverse diseases [21,22,23]. Abnormal expression of different RHO GTPases has been reported in many human tumors [24]. While most studies on RHO GTPase-associated tumorigenesis have narrowly focused on cytoskeletal-related biological processes and canonical pathways, the contribution of noncanonical mechanisms has also been observed [16]. These mechanisms include the regulation of autocrine and paracrine loops critical for remodeling of the tumor microenvironment and tumor growth [25], nucleolar functions connected with either efficient ribogenesis or the suppression of nucleolar stress in cancer cells [26,27,28], the regulation of both the centrosome and chromosome stability [27,29], the regulation of the YAP/TAZ pathway [30], and the recruitment of pathways to avoid antitumoral immune responses [31]. These observations highlight the key role of RHO GTPases in tumorigenesis but, at the same time, challenge the widely established functional archetypes in the field and the therapeutic feasibility of these pathways.

Contrary to recent studies that report the identification of new driver mutations in some RHO GTPases members, such as RAC1, RHOA, and CDC42 [32], analysis of cancer genomes has demonstrated that mutations affecting RHO signaling pathways are found at low frequency in a limited spectrum of tumor types [18,21,33]. While mutations in genes encoding RHO GTPases are rare in cancer, alternative mechanisms lead to the spurious activation of RHO pathways, which could add multiple regulatory layers controlling the steady-state levels and activation dynamics of RHO signaling elements. Consistent with this view, RHO GTPases can be further regulated by several approaches, including transcriptional regulation [34], alternative splicing [33,35,36], microRNA (miRNA)-mediated transcript stability [37], protein steady-state levels, posttranslational modifications [38], sequestration in endosomes [12], time of residence in membranes [39], intracellular trafficking [40], and F-actin-dependent cytoskeletal events [41]. In particular, miRNAs, as negative regulators of gene expression, repress RHO GTPases through degradation or translational blockade of their associated mRNAs [42,43]. While the processing of mRNA by miRNAs is essential for gene expression, the established role of long noncoding RNAs (lncRNAs) in the regulation of miRNAs highlights the potential mechanisms by which these RNA subtypes contribute to neoplastic transformations [44,45].

The involvement of lncRNAs in tumorigenesis and cancer progression may result from their roles in cell division, migration, differentiation, and apoptosis [46,47]. As critical gene regulators, lncRNAs interact with DNA, RNA, and proteins to modulate gene expression at the pretranscriptional, transcriptional, and posttranscriptional levels [48,49,50,51]. In this regard, the associations between lncRNA and the regulatory machinery of chromatin remodeling, transcription, splicing, and nuclear trafficking clarify the details of the regulatory aspects of these RNA species [52,53,54]. lncRNAs act as competing endogenous RNAs (ceRNAs) to regulate other RNA transcripts by competing for shared miRNAs [52,55]. According to the dysregulation of many miRNAs in RHO GTPase-dependent malignancies, lncRNAs are, as negative regulators of miRNAs, likely to become a focus in the future development of new RHO GTPase-based therapies [56]. Significantly, recent studies have reported that lncRNAs and members of RHO GTPase signaling cascades are dysregulated in various human cancers (Figure 2) [53,54].

miRNAs are noncoding RNA molecules that can control the expression of their target mRNAs [57]. These short sequences suppress target genes by either inhibiting translation or initiating degradation of mRNA based on complementarity [58]. The expression of several RHO GTPases can be regulated by miRNAs [59]. Notably, most of the studies on miRNAs have been performed in cancer models, and demonstrated that the modulation of RHO GTPase expression with miRNAs can affect cancer development. For instance, it has been reported that RHOA is downregulated by miR-340-5p and mir-200 to inhibit cancer progression [60,61]. RHOB is the target of miR-21, and RHOBTB1 is suppressed by miR-31 [62,63]. Also, CDC42 is a target of miR-29 and miR-137 [64,65]. Given the significance of miRNAs in the direct regulation of members of the RHO GTPase family, emerging evidence has revealed that numerous lncRNAs that modify RHO GTPases are likely to act as ceRNAs. These lncRNAs function as oncogenes by sponging tumor suppressor RHO GTPase-related miRNAs (Figure 2) [66,67], thereby indirectly contributing to the regulation of the expression of the RHO GTPase genes targeted by these miRNAs. Thus, the current knowledge of RHO GTPase-related oncogenic lncRNAs in relation to their aberrant expression and their mechanism of action through sponging effects on RHO GTPase-targeting miRNAs are discussed.

Up to now, a number of lncRNAs related to a few RHO GTPase members, such as RHOA, RHOB, RHOC, RAC1, and CDC42, have been identified. Therefore, this article addresses the current state of knowledge about the regulatory mechanisms of lncRNAs in the expression of genes related to RHOA, RHOB, RHOC, RAC1, and CDC42 with a specific focus on lncRNAs as molecular sponges of miRNAs.

### 1.1. RHOA-Related lncRNAs

Metastasis-associated lung adenocarcinoma transcript 1 (MALAT1) is a well-studied lncRNA. It is extremely abundant in nuclear speckles and was originally discovered in a study of metastasis in patients with early-stage NSCLC [68,69]. MALAT1 is located at chr11q13 and is approximately 8500 nucleotides in length. It is conserved throughout mammalian species, and plays important roles in development and evolution [70]. Although MALAT1 is abundant in normal cells, many studies have shown that this lncRNA promotes cell proliferation, migration, and metastasis in cancer tissues by affecting several signaling pathways, such as Wnt/β-catenin [71] and PI3K/AKT [71]. MALAT1 has molecular functions in the alternative splicing of pre-mRNA and in transcriptional and posttranscriptional regulation [72]. One of the main mechanisms of MALAT1 in posttranslational regulation is its function as a ceRNA that leads to the progression of various cancers [73]. Tumor-suppressor activity of miR-429 in breast cancer [74], CRC [75], and nasopharyngeal cancer [76] has been approved by different studies. Xiao et al. reported that lncRNA MALAT1 had high expression in human lung adenocarcinoma tissues (vs. paracancerous normal tissues, *t* = 16.387, *p* < 0.001), and its expression was associated with tumor size, lymph node metastasis, and TNM staging in patients. In this study, thirty-nine cases of lung adenocarcinoma tumor tissues and normal tissues from 22 males and 17 females, aged 45–78 years, were collected and the two-year survival of patients with low expression of MALAT1 was significantly higher than that of MALAT1 high expression in lung adenocarcinoma patients (Long rank = 4.773, *p* = 0.0289) [77]. MALAT1 acts as a ceRNA for miR-429 to regulate RHOA expression in lung adenocarcinoma cells. The upregulation of MALAT1 promoted cell proliferation, invasion, and migration of lung cancer cells by sponging miR-429 and consequently inducing RHOA expression [77].

Noncoding RNA activated by DNA damage (NORAD; also referred to as LINC00657) is a cytoplasmic lncRNA transcribed from the exon of chr20q11.23 [78]. This highly conserved lncRNA is upregulated in response to DNA damage, and it is required for genome stability as well as proper mitotic divisions through sequestration and negative regulation of PUMILIO (PUM) proteins, which are among deeply conserved RBPs negatively regulating gene expression [79,80]. NORAD has a crucial function in carcinogenesis, and its dysregulation has been implicated in various types of cancers [81,82,83]. Several studies demonstrated that NORAD could act as a ceRNA to sponge miRNAs and ultimately regulate the expression of several factors that play vital roles in cancer progression [84,85,86,87]. The ability of miR-125a-3p to increase the chemosensitivity of pancreatic cancer cells by inhibition of epithelial-to-mesenchymal transition (EMT) was reported [88]. Li et al. have shown that NORAD promoted the migration and invasion of pancreatic cancer cells through competitive binding to miR-125a-3p, causing RHOA deregulation [89]. In this study, analysis of microarray expression profiles between 55 pancreatic ductal adenocarcinoma cells (PDAC) and normal pancreas tissues, showed the significant upregulation of NORAD in cancer patients (*p* < 0.001). They also suggested that the upregulation of NORAD could occur through hypoxia, which thereby stimulates hypoxia-induced EMT [89].

The role of lncRNA zinc finger antisense 1 (ZFAS1) in breast cancer was originally exhibited by Askarian-Amiri et al. [90]. ZFAS1, a 17,561 bp transcript from the 5′ end of the ZNFX1 gene, is located on chr20q13.13. The expression of ZFAS1 is low in breast cancer cells, and it has antitumor activity [90,91]. Numerous studies have revealed its overexpression and roles in tumor growth and metastasis in various cancers, including bladder cancer [92], hepatocellular carcinoma (HCC) [93], colorectal cancer (CRC) [94], prostate cancer [95], and NSCLC [96]. In fact, being a molecular sponge for miRNAs is one of the regulatory mechanisms whereby ZFAS1 serves to adjust the proliferation, invasion, and metastasis of cancer cells [97]. Liu et al. investigated the differential expression of ZFAS1 using nine GEO datasets (252 tumor samples and 56 normal samples), and identified that ZFAS1 had upregulated expression in PDAC (adjusted *p*. value (3.79E−10) [44]. ZFAS1 expression level in PDAC were further validated by data from the ONCOMINE, UALCAN, and GEPIA databases; all of them supported the GEO results, and it was also recognized to act as a miR-3924 sponge [44]. It was claimed that ROCK2 can be considered as the target of ZFAS1/miR-3924, and its level of expression was decreased upon ZFAS1 knockdown. Therefore, the contribution of ZFAS1/miR-3924 in tumor metastasis via the RHOA/ROCK2 pathway was reported, confirming its potential regulatory effect on RHO GTPase members [44].

Prostate cancer-associated transcript 6 (PCAT6) is a member of a group of 121 lncRNAs associated with prostate cancer [98]. PCAT6 is an intergenic lncRNA that is located at chr1q32.1, and it is reported to have an oncogenic role in various human cancers [99]. Several studies demonstrated that PCAT6 was overexpressed in several tumor tissues, such as GC [100], HCC [101], cervical cancer [102], NSCLC [103], lung cancer [104], and CRC [105], and this overexpression could facilitate the proliferation and metastasis of those cancers. It was indicated that PCAT6 may act as a ceRNA to suppress miRNAs and regulate the expression of their targets [99]. Tu et al. reported that the relative PCAT6 expression levels in CCA specimens were obtained compared with non-tumor tissues (*n* = 20 pairs). This was owing to the study of the relationship between PCAT6 expression and the development of CCA. High expression of PCAT6 was observed in CCA tissues compared to the non-tumorous tissues (*p* < 0.001). It was also found that PCAT6 induced M2 polarization of macrophages in cholangiocarcinoma via competitive binding to miR-326, and modulated the RHOA signaling pathway [45]. Low expression of miR-326 in many tumors was dramatically related with unfavorable prognosis, tumor development, metastasis, and progression [106]. Thus, PCAT6 may be a potential target of immunotherapy in cholangiocarcinoma treatment [45].

As a type of new intergenic lncRNA, LINC02381 is located at chr12q13.13 [107]. LINC02381 has been reported to be involved in many cancers, including CRC [107] and GC [108], as well as autoimmune diseases, such as rheumatoid arthritis [109]. LINC02381 can regulate the PI3K-AKT, MAP2K3/p38, and WNT signaling pathways via competitive binding to miRNAs, such as miR-96, miR-18, miR-21, miR-27a, and miR-590p [107,108,109]. The important role of miR-133b in the inhibition of tumorigenesis suggested its potential application as tumor-suppressor miRNA [110,111]. In the study by Chen et al., thirty groups of cervical tumor tissues and normal tissues were taken to illustrate that the expression of LINC02381 was remarkably upregulated in cervical cancer tissues (*p* < 0.05). In addition, patients with higher expression levels of LINC02381 had advanced tumor stage and lymph nodes metastasis [112]. Data showed that LINC02381 promoted cell viability and migration in cervical cancer cells through endogenous competition with miR-133b, which led to the overexpression of its downstream protein, RHOA. Therefore, LINC02381 is considered a novel target for the treatment of cervical cancer [112].

Prostate cancer gene expression marker 1 (PCGEM1) is a prominent lncRNA that has critical roles in the development and carcinogenesis of prostate cells [113]. PCGEM1 is mapped to chr2q32, and plays an important role in the apoptotic response, proliferation, colony formation, and progression of several types of cancers [114]. PCGEM1 acts as an oncogenic factor in cervical carcinoma [115], NSCLC [116], prostate cancer [113], and glioma [117] by acting as a ceRNA for miRNAs, such as miR-642a-5p, miR-433-3p, miR148a, and miR-539-5p. Results of the study showed that the expression level of PCGEM1 was significantly higher in ovarian cancer tissues than in normal ovarian tissues (14 normal ovarian tissue and 50 epithelial ovarian cancer tissue specimens; *p* < 0.05) [118]. Furthermore, PCGEM1 induced cell proliferation, migration, and invasion by targeting miR-129-5p and modulating the expression of RHOA and its downstream effectors [118]. Alternatively, the ability of miR-129-5p in the inhibition of ovarian cancer cell proliferation and survival was defined previously, which highlighted the role of the PCGEM1 and miR-129-5p as the treatment options in epithelial ovarian cancer [119].

Taken together, a list of the currently validated lncRNAs which are responsible for sponging RHOA-targeting miRNAs could act as a reference resource. This represents an extra layer of the complexity of the underlying mechanisms involved in RHO-driven cancers, and provides a promising avenue for therapeutic intervention.

### 1.2. RHOB-Related lncRNAs

Several studies have shown that RHOA and RHOC positively contribute to tumor promotion, but the distinct role of RHOB as a tumor suppressor or an oncogene in cancer remains elusive [120]. Additionally, in contrast to RHOA and RHOC, RHOB localizes not only at the plasma membrane but also on endosomes, multivesicular bodies, and in the nucleus, which could contribute to its contrasting behaviors in cancer biology [120]. Some studies have indicated that the oncogenic role of RHOB in tumor formation acts by inducing proliferation, angiogenesis, invasion, and migration [121,122,123]. On the other hand, more recent investigations have confirmed the downregulation of RHOB in some tumors, suggesting it has tumor suppressor activity [124,125]. lncRNA growth arrest specific 5 (GAS5), located at chr1q25.1, has been reported to have tumor suppressive activity in multiple human cancers by modulating several cellular processes [126]. GAS5 can act as a ceRNA to modulate the expression of many genes and their associated signaling pathways. Bioinformatics analysis of complementary regions of GAS5 to miRNAs determined 690 candidates, among which 234 miRNAs presented statistically significant binding [127]. GAS5 suppresses miR-21, an oncogene in numerous solid tumors and lymphoma, and FGF1, a regulator of proliferation and apoptosis, so it can be considered as a mediator of the GAS5/miR-21 axis [128]. In NSCLC, inhibition of GAS5 expression resulted in chemoresistance due to the competition between GAS5 and PTEN for miR-21 binding, regarding the tumor-suppressor activity of PTEN as a negative regulator of the AKT/PKB signaling pathway [129]. Furthermore, the capacity of GAS5 to bind to other oncogenic miRNAs confirms the pivotal role of this lncRNA in the modulation of different cancer-related genes [130,131,132]. Strikingly, regulation of RHOB by GAS5 illustrates the ability of this RNA molecule to be a modulator of the RHO GTPase family [53]. It was discovered that miR-663a expression was remarkably elevated in both tissues and plasmas of osteosarcoma patients [133]. In this regard, the GAS5 expression level was significantly reduced (*p* < 0.001) in osteosarcoma tissues (*n* = 20) compared with normal tissues (*n* = 20), and was negatively correlated with miR-663a expression. Moreover, it was found that RHOB expression can be negatively modulated by miR-663a. Therefore, upregulation of GAS5 and RHOB inhibited osteosarcoma cell proliferation, migration, and invasion in vitro, while overexpression of miR-663a induced malignancy in these cells [53].

Overall, the wide range of potential interactions between miRNAs and RHO GTPases, and lncRNAs such as GAS5 in various malignancies, provides new challenges and opportunities for the development of new treatment options (Figure 2).

### 1.3. RHOC-Related lncRNAs

RHOC has been widely reported to act as a master regulator of actin organization [134]. It has been shown to impact the motility of cancer cells, is essentially involved in invasion and metastasis, and contributes to carcinoma promotion in the breasts, pancreas, lungs, ovaries, and cervix, among several others [135,136]. The most interesting discovery has been its significant role in metastasis [137]. In addition, it has the ability to regulate various other features, such as cell migration, adhesion, cell polarity, angiogenesis, motility, invasion, metastasis, and anoikis resistance [138,139]. These findings suggest that RHOC contributes to the plasticity required for cancer cells to display diverse functions based on microenvironmental cues. Taken together, studies have also suggested that the inhibition of RHOC through gene regulators such as lncRNAs may be a new therapeutic target to abolish advanced tumor phenotypes.

Testis developmental related gene 1 (TDRG1) was first identified as a human testicular-specific gene, and proved to be a key regulator in reproductive organ-related cancers [140]. This lncRNA is a 1.1 kb transcript located at chr6p212.1-p21.2 and spans 1.18 kb with two exons and one intron. It encodes a 100 amino acid protein with unknown function [140,141]. TDRG1 was initially considered a key modulator in spermatogenesis and sperm motility [142]. Additionally, it may be involved in the promotion of testicular germ cell tumors [143]. Furthermore, as a modulator in reproductive organ-related tumors, TDRG1 was indicated to play a role in cell proliferation, migration, and invasion in endometrial carcinoma [141,144] and epithelial ovarian carcinoma. It was recently reported that TDRG1 is significantly overexpressed in bone marrow mesenchymal stem cells [145]. TDRG1 was found to be more highly expressed in (EOC) tissues than in normal ovarian tissues (*p*  <  0.05), and its downregulation inhibited EOC cell division, migration, and division [146]. Studies have indicated that TDRG1 is a potential binding site for miR-93, and upregulation of TDRG1 represses miR-93 expression [146]. Moreover, downregulation of TDRG1 decreased the expression of RHOC, p70S6K, BCL-XL, and MMP2, which are targets of miR-93. Other data indicated that serum levels of miR-93 (*p* = 0.0001) was downregulated in EOC patients compared with healthy women [147]. Taken together, these results demonstrate that TDRG1 could be an alternative therapeutic target in EOC or other cancers through the inhibition of RHOC [146].

Among the characterized biomarkers, prostate cancer gene 3 (PCA3) is considered one of the most promising for its diagnostic potential for the prediction of prostate biopsy results and therapeutic outcomes [148]. PCA3 (also known as DD3 or DD3PCA3) is located on chr9, and is transcribed into a noncoding prostate-specific RNA that is upregulated 60 to 100 times in tumor cells compared to normal prostate tissue [149]. The results of another study showed that the expression level of lncRNA PCA3 in epithelial ovarian cancer cell lines including BeWo cells, JEG-3 cells, and JAR cells was higher than that in normal ovarian HTR-8 cells (*p* < 0.001) [150]. It was discovered that suppression of PCA3 expression in EOC cells by siRNA transfection remarkably impeded cell division, migration, and invasion [150]. While extensive exploration of miR-106b-5p has determined opposite functions involved in different cancer cell biological behaviors, its tumor-suppressor activity in epithelial ovarian cancer tissues was observed [151]. Bioinformatics analysis and dual-luciferase reporter assays reported that PCA3 had potential binding sites for miR-106b-5p. Thus, PCA3 could function as a molecular sponge and prevent miR-106b-5p from binding to its targets. Knockdown of lncRNA PCA3 by siRNA resulted in overexpression of miR-106b and downregulation of RAS, RHOC, BCL/XL, p70S6K, and MMP2, which are targets of miR-106b [150].

Recognition of TDRG1 and PCA3 as the two positive regulators of RHOC with the ever-increasing number of other RHO GTPase-specific lncRNAs strongly indicates their potential contribution with the entire process of *RHO GTPase*-driven carcinogenesis (Figure 2).

### 1.4. RAC1-Related lncRNAs

RAC1 plays diverse roles in dynamic cell biological processes, including cell survival, cell-cell contacts, cell motility, EMT, and cell invasiveness [152]. Proper function of RAC1 is provided through precise regulation by regulators and downstream effectors and their modifications to its specific accumulation and subcellular localization [2,24]. Hence, any disturbances in each step of regulation produce critical changes to the expression and activity of RAC1, which in many cases can lead to cancer progression and metastasis. To date, overexpression of RAC1 has been observed in breast cancer [153], while its mutations have been identified in prostate [154], testicular [155], melanoma [156], and NSCLC cancers [157]. Hence, it is obvious that the number of RAC1 mutations and the mechanism of perturbations in RAC1 expression should be clearly studied to design new drugs for cancer patients. One of the regulatory mechanisms involved in the regulation of RAC1 expression is the sponging effect of lncRNAs, including CDKN2B-AS1, AURKAPS1, LCAT1, LSINCT5, TP73-AS1, FTH1P3, and XIST (Figure 2).

Cyclin-dependent kinase inhibitor 2B antisense RNA 1 (CDKN2B-AS1), also known as ANRIL, is located at the INK4 locus in the antisense direction of the CDKN2A and CDKN2B genes, and has epigenetic regulation of these nearby genes by recruiting the polycomb repressor complex 1 and 2 [158,159]. CDKN2B-AS1 is an oncogenic lncRNA that is upregulated in many cancerous tissues, such as ovarian cancer [160], bladder cancer [161], laryngeal squamous cell cancer [162], HCC [163], cervical cancer [164], and NSCLC [165]. The overexpression of CDKN2B-AS1 correlated with cell invasion, proliferation, tumor metastasis, and inhibition of apoptosis and senescence [166]. Various articles have shown that CDKN2B-AS1 could interact with miRNAs, such as miR-181a-5p [166], let-7c-5p [167], miR-143-3p [160], miR-122 [168], and miR-378b [169], to regulate cancer progression. Dasgupta et al. found that CDKN2B-AS1 expression as an oncogenic lncRNA increased from lower grade and stage to higher grade and stage in renal cell carcinoma based on the TCGA-KIRC (normal  =  72, tumor  =  518), TCGA-KICH (normal  =  25, tumor  =  66), and TCGA-KIRP (normal  =  30, tumor  =  197; *p* < 0.05) databases. Moreover, higher expression was significantly (*p*  <  0.0001) correlated to overall survival. CDKN2B-AS1 targeted miR-141 to induce tumor progression and metastasis in renal cell carcinoma via the cyclin D/RAC1/paxillin network. Strikingly, cyclin D1 and D2 were direct targets of miR-141, which could regulate RAC1 and phospho-paxillin expression and contribute to EMT [170].

AURKAPS1 is the pseudogene of AURKA (Aurora Kinase A), and it is a protein-coding gene located on chr1 in the intron of the RAB3GAP2 gene. Compared with the AURKA gene, AURKAPS1 has the absence of a 359–560 coding sequence region, and differences in the 3′-terminal nucleotide sequence and nucleotide mutations. To date, the function and abnormal expression of AURKAPS1 in cancerous tissues have been reported in few articles [171]. Li et al. reported that AURKAPS1 expression in 124 cases of HCC tissues was significantly higher than in adjacent normal liver tissues. Additionally, the expression level of AURKAPS1 was positively correlated with tumor size (*p* < 0.006) and TNM stage (*p* < 0.046), confirming its carcinogenic role [171]. AURKAPS1 increased HCC cell migration and invasion through competitive binding to miR-142, miR-155, and miR-182, promoting the expression of their target RAC1. Although the high expression of miR-182-5p and miR-155 correlated with poor prognosis and acceleration of liver cancer cell growth, the inhibition of metastasis for miR-142 has been reported in HCC [171,172,173]. Hence, AURKAPS1 and its corresponding miRNAs could be considered appropriate cancer biomarkers to provide a promising therapeutic strategy for human liver cancer [171].

For the first time, Yang et al. introduced lung cancer-associated transcript 1 (LCAT1) as a novel lncRNA that does not have protein-coding capacity [174,175]. LCAT1, located at chr2q31.1, is 896 bp in length and contains one transcript with three exons. Analysis of RNA-seq data of 485 lung adenocarcinoma tissues and 56 adjacent normal tissues from TCGA, and qPCR results of 25 paired lung adenocarcinoma tissues and corresponding adjacent normal tissues was performed. Data revealed that LCAT1 expression was elevated in lung cancer tissues in comparison to normal tissues (*p* <  0.001), so it could promote cell proliferation, migration, and invasion [175]. In these cells, LCAT1 indirectly upregulated RAC1 and its downstream effector, PAK1, through competitive binding to miR-4715-5p [175]. Similarly, it was found that HOTAIR as a ceRNA for miR-4715-5p promoted the cell growth, migration, and invasion [176]. Accordingly, the LCAT1-miR-4715-5p-RAC1/PAK1 axis could be a specific target for the treatment of lung cancer patients with LCAT1 overexpression in their cancer cells [174].

LSINCT5 (long stress-induced noncoding transcript) is an intergenic lncRNA located at chr5:2,765,705–2,768,351 between the IRX4 and IRX2 genes. LSINCT5 is a 2.6-kb polyadenylated transcript, and is transcribed from the negative strand by RNA Polymerase III [177]. Numerous studies have shown that LSINCT5 could be implicated in cancer development and has critical roles in metastasis, proliferation, and apoptosis of cancerous cells. Bladder cancer [178], osteosarcoma [179], endometrial carcinoma [180], gastrointestinal cancer [181], and breast and ovarian cancers [177] are among the cancers in which LSINCT5 plays significant roles in their progression. Liu et al. demonstrated that LSINCT5 expression in 56 glioma tissues was remarkably higher than that in 16 normal samples (*p*  < 0.001) [182]. Knockdown of lncRNA LSINCT5 in human glioma cells triggered cell apoptosis and suppressed cell viability, migration, and invasion. LSINCT5 knockdown led to miR-451 overexpression and RAC1 downregulation, inhibiting the PI3K/AKT, WNT/β-catenin, and NFκB pathways [182]. Strikingly, downregulation of miR-451 in gliomas has been suggested by several different research groups with the role in suppression of cell growth, proliferation, and induction of cell apoptosis [183].

p73 antisense RNA 1 (TP73-AS1), also known as KIAA0495 or PDAM, is transcribed from chr1p36, and has an approximately 216 bp overlap with the TP71 gene located on the opposite strand [184]. Many studies have revealed that TP73-AS1 has a critical role in the development of different cancers [185]. Tumor-node-metastasis stage, tumor size, lymph node metastasis, and prognosis are among the clinicopathological characteristics associated with abnormal TP73-AS1 expression. Moreover, TP73-AS1 could participate in the promotion of cancer cell proliferation, invasion, and metastasis as well as the inhibition of apoptosis [184]. In some cancers, this lncRNA could act as a ceRNA to prevent the degradation of mRNA [186,187,188,189,190]. For example, Yang et al. reported the significant higher expression of TP73-AS1 in 46 primary osteosarcoma tissues in comparison with their matched adjacent normal bone (*p* < 0.05). In this study, the overexpression of TP73-AS1 could facilitate osteosarcoma cell proliferation and invasion in vitro, as well as tumor growth in vivo through competitive binding to miR-142, which could thereby positively regulate RAC1 protein [191]. Moreover, the significant effect of miR-142 on suppression of proliferation and induction of apoptosis for osteosarcoma cells has been recognized [192].

FTH1P3 (ferritin heavy chain 1 pseudogene 3, NR_002201) is considered a member of the FHC gene family. It is approximately 954 nucleotides in length, and located on chr2p23.3 [193]. FTH1P3 is a newly identified lncRNA in cancer cells, and its overexpression in cancer cell lines and tissues has been demonstrated by several studies. FTH1P3 can promote cell proliferation, migration, invasion, and tumor progression in uveal melanoma [194], oral squamous cell carcinoma [193], laryngeal squamous cell carcinoma [195], cervical cancer [196], and NSCLC. Zheng et al. reported that the expression level of FTH1P3 was upregulated in uveal melanoma cell lines (C918, MUM-2B, OCM-1A, and MUM-2C) compared to that melanocyte cell line (D78). Similarly, in 25 uveal melanoma patient samples, a higher expression of FTH1P3 was observed than in the 25 normal samples (*p* < 0.01) [194]. In addition, it was indicated that the upregulation of FTH1P3 targeted miR-224-5p in uveal melanoma, and caused an increase in the expression of RAC1 and Fizzled 5, which are direct target genes of miR-224-5p. Particularly, other studies proved that miR-224-5 contributed to the proliferation, invasion, and migration of uveal melanoma cells by regulating the expression of PIK3R3 and AKT3 [197]. As a result, FTH1P3 has a critical role in uveal melanoma progression, and could be a potential therapeutic target for uveal melanoma [194].

One of the first lncRNAs identified to be involved in the epigenetic processes was X-inactive–specific transcript (XIST). The XIST gene is located in the X chromosome inactivation center of chrXq13.2, and was discovered in early 1990 [198]. XIST has an indispensable role in the X-chromosome inactivation (XCI) process for dosage compensation of sex chromosomes between males (XY) and females (XX) [199]. Despite its original role in XCI, XIST also participates in the regulation of cell growth and development as well as the progression of tumors and other human diseases, such as Alzheimer’s [200], acute myocardial infarction [201], cardiac morbidities [202], myocardial infarction [203], and acute kidney injury [204]. Many studies have demonstrated the aberrant expression of XIST in cancer cells and tissues, and revealed the association of XIST with tumorigenesis, metastasis, and cell apoptosis. Bladder [205], breast [206], colorectal [207], lung [208], melanoma [209], gastric [210], and ovarian cancers [211] as well as osteosarcoma [212] and glioma [213] are all among these cancers with aberrant expression of XIST. Data revealed that XIST expression was dramatically upregulated in 30 glioma tissues compared with 18 normal brain tissues, and was positively correlated with tumor grade (*p* < 0.05) [214]. Its function as a ceRNA is one of the most common mechanisms by which XIST contributes to cancer promotion [215]. For instance, in glioma, lncRNA XIST modulated RAC1 expression by functioning as a ceRNA of miR-137, which led to cell proliferation. While in glioma, the antiapoptotic capability of tumor cells has been associated with disease progression and therapy resistance, and it has been shown that miR-137 could induce apoptosis in cancer cells [216]. Thus, it was revealed that the XIST-miR-137-RAC1 pathway regulatory axis could be a therapeutic target in the treatment of glioma [214].

Collectively, these data suggest that dysregulation of network corresponding RAC1, miRNAs, and oncogenic lncRNAs needs to be carefully examined to increase our understanding about potential oncogenic drivers associated with the RHOC-related tumorigenesis (Figure 2).

### 1.5. CDC42-Related lncRNAs

CDC42, a prominent member of the RHO subfamily, is connected with cytoskeletal dynamics, cell shape, cell polarity, cell cycle progression, division, invasion, migration, and metastasis [19]. Its significant role as a regulator of numerous cellular processes indicates its involvement in the promotion of many human cancers [217,218]. CDC42 is connected with the tumorigenesis and promotion of esophageal squamous cell carcinoma (ESCC) [219,220]. Upregulation of CDC42 may induce HCC proliferation and metastasis, while miR-195 was shown to inhibit metastasis of HCC by suppressing CDC42 expression [221]. Moreover, it was reported that miR-29, as an invasion suppressor, downregulates CDC42 in gliomas [222]. It was also found that CDC42 could promote cellular invasion and metastasis in breast and colon cancers [223]. The ability of several lncRNAs to regulate RHO GTPases through their sponging activity has been reported. In this regard, MALAT1 is recognized as one of these lncRNAs because it increased the expression of CDC42 via miR-1 sponging [224]. ZFAS1 is another lncRNA considered to be a molecular sponge for miR-509-3p with a regulatory effect on CDC42 [225]. There are also other types of CDC42-related lncRNAs that are explained in the following sections (Figure 2). On the basis of the remarkable role of CDC42 in the processes underlying tumor formation, newly recognized regulatory mechanisms for CDC42 and CDC42-related signaling pathways, such as lncRNAs, may inhibit the process of neoplastic transformation by impeding the abnormal overexpression of CDC42.

H19 is an imprinted gene in the 11p15.5 region, and is expressed exclusively from the maternal allele. Overexpression of H19 in cancer cell lines and sample patients and its oncogenic activities have been demonstrated by several studies [226]. Functionally, H19 mainly acts as a molecular sponge or ceRNA of its diverse miRNA targets, such as miR-874, miR-675, miR-200, miR-107, miR194, miR-130a, miR196b, miR-193b, and let-7b, and it modulates the expression of many genes, resulting in multiple functions in the regulation of different cellular processes [227,228,229,230,231,232,233,234]. Thus, its role as the critical regulator of many targeted genes indicates its definite association with the development and progression of many human tumors [235,236]. For instance, overexpression of H19 in samples of primary HCCs (*n* = 46) and in comparison with the adjacent normal liver tissues (*n* = 46) was observed [237]. Strikingly, while H19 and CDC42 were overexpressed, miR-15b was downregulated in HCC cells and tissues [237]. The high expression of miR-15b-5p has been revealed to play an essential part in hepato-carcinogenesis through diverse regulation approaches [238]. H19 knockdown suppressed proliferation, migration, and invasion and increased apoptosis, which was rescued by a miR-15b inhibitor. In addition, H19 knockdown inhibited the CDC42/PAK1 pathway and EMT progression, providing a deeper understanding of the miR-15b/CDC42/PAK1 axis and the regulatory effects of H19 in HCC carcinogenesis [237].

Taurine upregulated 1 (TUG1) is a novel lncRNA with 7598 nucleotides. It is localized to chr22q12.2, and it is associated with tumorigenesis [239]. In recent years, TUG1 has been shown to be abnormally expressed in multiple types of cancers and plays crucial regulatory roles in various cancer-associated biological processes, such as the regulation of cell proliferation, apoptosis, differentiation, angiogenesis, invasion, metastasis, and drug resistance [239,240]. A recent publication showed that in 27 paired ESCC tissues and adjoining normal esophageal tissues, TUG1 was conspicuously upregulated in cancer samples (*p* < 0.0001), and high TUG1 expression was correlated with tumor size (*p* = 0.032), TNM stage (*p* < 0.001), and lymph node status (*p* < 0.001) [241]. Therefore, enhanced TUG1 expression might be related to the development of ESCC. This lncRNA exerts its function via several molecular mechanisms mainly by acting as a ceRNA to sponge various miRNAs. An increasing number of studies have demonstrated that the sponging activity of TUG1 as an oncogenic lncRNA has been implicated in many human malignancies [242,243,244]. Whereas overexpression of miR-498 in ESCC cell lines induced remarkable reductions of cell proliferation, the decrease in TUG1 reduced CDC42 expression by binding to miR-498, resulting in silencing of the proliferation and invasion of ESCC [245]. Importantly, the increase in proliferation and invasion induced by miR-498 reduction was improved by CDC42 overexpression, indicating that TUG1 could be a potential therapeutic target for ESCC [241].

The lncRNA small nucleolar RNA host gene 1 (SNHG1) is reported to increase cell proliferation, migration, and invasion in different cancers, including HCC [246,247], cervical cancer [248], prostate cancer [249], and non-small cell lung cancer [250]. Determination of the expression level in 36 esophageal cancer (EC) tissue samples and paracancerous tissues were obtained, revealing that SNHG1 was significantly upregulated in EC (*p*  <  0.01) [251]. However, few studies have evaluated the role of SNHG1 in cell migration and invasion in EC. It was reported that decreased expression of miR-195 played a regulatory role in promoting the pathogenesis of the EC [252]. While SNHG1 and CDC42 were significantly upregulated, miR-195 was remarkably repressed in both EC tissues and cell lines. Additionally, the suppression of either SNHG1 or CDC42 led to inhibition of cell proliferation, migration, and invasion. However, the inhibition of miR-195 resulted in the promotion of cell proliferation, migration, and invasion, and reversed the effects of si-SNHG1 [251].

Small nucleolar RNA host gene 15 (SNHG15) is a typical lncRNA that has been revealed to be upregulated as a tumor facilitator in NSCLC [253], CRC [254,255], breast cancer [256], pancreatic cancer [257], and GC [258]. Mechanistic investigation found that SNHG15, as the molecular sponge of miR-200a-3P, upregulated YAP1 as an oncogene and led to induction of the Hippo signaling pathway [259]. The results of another study indicated that SNHG15 is highly expressed in glioma vascular endothelial cells (*p* < 0.05), while its knockdown suppressed cell proliferation, migration, and tube formation in vitro [260]. Moreover, knockdown of SNHG15 reduced the expression of VEGFA and CDC42, which are promoters of angiogenesis. Additional analysis confirmed that SNHG15 affected endothelial cell function by targeting miR-153 as a negative regulator of CDC42 and VEGFA. The tumor-suppressor activity of miR-153 was determined in another study through the reduction of stem cell-like phenotype and tumor growth of lung adenocarcinoma cells [261]. Therefore, SNHG15 and its target miR-153 could serve as new potential therapeutic targets for the antiangiogenic treatment of glioma through the downregulation of CDC42 [260].

Data summarized here demonstrate a strong rationale for targeting CDC42-related regulator lncRNAs as a potential therapeutic approach to control various cancer-related biochemical processes (Figure 2).

## 2. Indirect Regulatory Effects of lncRNAs on RHO GTPases

The on and off states of the RHO GTPases is directly regulated by three specific classes of regulator families, consisting of multiple different members (Figure 1) [2]. Furthermore, the capacity of RHO GTPases to mediate a wide range of intracellular signaling pathways is attributed to their connection with their diverse downstream targets, called effector proteins [2]. To date, more than 70 potential effectors as the kinases or scaffolding proteins have been identified for RHO GTPases [262,263]. Kinases are recognized as an important class of RHO effectors, and result in downstream phosphorylation signal transductions [264]. The second group of effectors comprise of scaffolding proteins, which probably form a framework contributed in signaling cascades, particularly through reorganization of filamentous actin dynamics [265,266].

In addition to direct regulation of Rho GTPases themselves, lncRNAs as ceRNAs are also indirectly involved in modulation of RHO GTPases by targeting their regulators (GAPs, GEFs, and GDIs) and effectors. Upregulation of CTC-497E21.4 (also known as LINC00958) in GC tissue promoted cell proliferation, invasion, and metastasis by sponging miR-22, and subsequently induced the expression of NET1 [41]. Interestingly, NET1 activates as a RHOA-specific GEF RHOA signaling pathway. Thus, CTC-497E21.4/miR-22/NET1 can be referred to as an indirect modulatory axis of RHOA-mediated signaling [41]. Rho-associated coiled-coil containing kinases (ROCKI and ROCKII) act as RHOA effectors in the regulation of cellular contraction, motility, morphology, polarity, gene expression, and cell division [267]. An elevated expression of MALAT1 was observed in osteosarcoma patients, and correlated with poor prognosis. Downregulation of MALAT1 as a ceRNA for miR-144-3p inhibited tumor cell invasion by reducing the expression of ROCKI/ROCKII in osteosarcoma cells [268]. The modulatory effects of lncRNAs on RHO GTPase regulation and signaling adds another level of complexity and stringency to existing regulatory frameworks and control mechanisms that require further investigation.

## 3. Epigenetic Regulation of RHO GTPases by lncRNAs

It is now clear that perturbations of epigenetic regulation are a key feature of many neoplastic transformations [269]. In the nucleus, different lncRNAs play a vital role as the key regulators of the epigenetic status of protein-coding genes. In fact, of the diverse array of functions assigned to lncRNAs, one attractive perspective is the direct interaction between lncRNAs and epigenetic modifiers to modulate chromatin conformation [270]. Recruitment histone-modifying machineries, such as writers, readers, and erasers of histones to the specific subnuclear domains and genetic loci, result in the fine-tuning of chromatin structure [271]. Many researches have established that EZH2 functions as an inhibitor of RNA transcription by histone modification; namely, H3K27me3. It was found that TUG1 represses the expression of *RND3,* one atypical member of the RHO family [14,36], through recruitment of EZH2 protein to the *RND3* promoter regions and modification of H3K27me3 [239]. Based on another paradigm, lncRNAs could be involved in modification of nuclear architecture and the three-dimensional genome structure. Therefore, orchestrating chromatin folding and compartmentalization to direct enhancer-promoter communication shapes the outcomes of gene transcription [272]. Another mechanistic layer of lncRNAs-mediated epigenetic alternations is associated with alternative splicing. Interaction between lncRNAs and splice-regulatory factors may influence the processing of pre-mRNA, and thus of selecting the alternative transcript isoforms [273]. Interestingly, lncRNAs are not only recognized as the master regulators of the genome epigenetics, but also as the targets of epigenetic modifiers [274]. It has been demonstrated that modified transcripts of lncRNAs, such as their N6-Methyladenosine (m6A) residues, are associated with cancer progression [275]. For instance, m6A-modified transcripts of GAS5, as one of the regulators of RHOB, induced phosphorylation and subsequently ubiquitin-mediated degradation of YAP, leading to the inhibition of malignant transformation [276]. m6A reader YTHDF3 facilitates the degradation of m6A-modified lncRNA GAS5 and thus contributes to cancer development. In conclusion, lncRNAs act as regulators and targets of epigenetic factors, which establishes a cross-regulating network in tumors and unveils a novel dimension of cancer biology.

## 4. lncRNAs as Novel Therapeutic Targets

The well-established role of the lncRNAs in different diseases provides a theoretical basis to utilize lncRNAs as pharmacological targets. Recent progress in the generation, purification, and cellular delivery of RNAs have enabled the development of RNA-based therapeutics, including small interfering RNAs (siRNAs), antisense oligonucleotides (ASO), aptamers, microRNAs, and messenger RNAs. SiRNAs complementary to the target lncRNAs recruit the RNA-induced silencing complex (RISC) and induce lncRNA degradation [277,278,279,280]. Antisense oligonucleotides, which are chemically synthesized short single-stranded DNA sequences with 15–20 nucleotides, can bind to complementary RNA and provoking RNA degradation or inhibit translation [281]. Application of specific ASO technology, including peptide nucleic acids (PNAs), phosphorodiamidate morpholino oligomers, and locked nucleic acids (LNA) for lncRNA-targeted therapy has grown rapidly, indicating their clinical prospects [282,283]. Particularly, the most recent third generation ASOs, including PNAs with modified nucleotides, are often used for targeting lncRNAs [284,285]. According to specific cellular localization of lncRNAs in the nucleus or the cytoplasm, some therapeutic strategies could be applied for more effective lncRNAs targeting. In this regard, ASOs and siRNAs can more effectively inhibit nuclear and cytoplasmic lncRNAs, respectively. More efficiently, for lncRNAs with dual cellular or uncertain localization, a combination targeting strategy using both ASO and siRNA has been shown to be successful [286,287]. The CRISPR/Cas9 system as an alternative RNA-based therapy with successful results has stimulated research into lncRNA-targeting [288,289]. In addition to nucleic acid-based therapeutic technologies, small-molecule compounds have been proven to be effective entities to disrupt lncRNA spatial secondary and tertiary structures or lncRNA-protein interactions [290,291]. Recently, a number of studies have demonstrated that plant-derived natural compounds have a credible regulatory effect on lncRNAs; however, the lack of accurate targets and mechanisms is a limitation of phytochemicals [292]. In the past few years, approval of ASO and siRNA-based therapies by the FDA have led to clinical breakthroughs [293]. Primary results are conceivable that targeting lncRNAs provides a plethora of opportunities in the near future for precision medicine. For instance, studies into mitochondrial lncRNAs (mtlncRNAs) have progressed rapidly, and to this end, the FDA has approved a clinical trial of Andes-1537, a short single-stranded phosphorothioate ASO. This type of ASO binds to the antisense non-coding mitochondrial RNA (ASncmtRNA), and it is under clinical investigation for the targeted therapy of solid tumors, including a phase I clinical trial of the patients with advanced metastatic cancer (NCT02508441). The clinical results of phase I confirmed that Andes-1537 was well-tolerated. In addition, another clinical trial of Andes-1537 for different solid tumors (NCT03985072) was started last year [294]. Taken together, the application of lncRNAs as pharmacological targets has reviewed the transition of lncRNAs from the role of disease coding to acting as drug candidates, providing new insights into the treatment of diseases. Another therapeutic strategy is based on enzymatic DNA (DNAzymes), which exploits the catalytic capabilities of DNAzymes to cleave target RNAs, such as mRNAs and lncRNAs, with exceptional selectivity [295]. In contrast to gene-editing approaches, such as CRISPR/Cas9, DNAzymes do not permanently modify the genome, and enable a transient and dose-dependent reduction of the target-RNA levels. In contrast to miRNA, siRNA, or ASO, DNAzymes are self-sufficient catalysts that do not depend on cellular proteins. This strategy might provide eminent advantages over existing approaches, including effective responses to emerging drug resistances and the fast development of personalized drugs.

## 5. Conclusions and Future Directions

The RHO family GTPases, as signal transduction elements, plays vital roles in several central biological processes that are remarkably essential for the maintenance of cellular homeostasis, and any perturbation of their signaling function causes diverse human diseases, especially cancer progression and metastasis [17,18,19,20]. Hence, comprehensive studies on different aspects of RHO GTPase structures, functions, and interactions as well as their mechanisms of regulation could help us to find new selective therapeutic strategies for cancer treatment. Among the various regulatory actions in live cells, lncRNAs orchestrate diverse fundamental processes, such as gene expression, epigenetic regulation, genomic imprinting, chromosome organization, allosteric regulation of enzyme activity, and miRNA sponging [296]. Determination of the modulatory mechanisms of lncRNAs concerning the RHO GTPase signaling pathways is anticipated to substantially expand our knowledge about the mechanisms of cancer progression. The role of lncRNAs as molecular sponges of miRNAs that target RHO GTPases should be further studied. It is worth mentioning that merely a limited assortment of well-known lncRNAs have been functionally characterized. However, growing evidence of their engagement in diverse human diseases and malignancies makes these transcripts versatile therapeutic targets for RNA-based therapeutic strategies [297]. In this review, we concentrated on the sponging effect as an approach for RHO GTPase downregulation. Since the majority of lncRNAs are expressed in a highly cell- or tissue-specific pattern, they may be efficient therapeutic targets for cancer therapy. Nevertheless, several questions remain to be addressed: (i) How many lncRNAs are functionally and clinically related to RHO GTPase-driven cancers? (ii) How can systematic genomic and functional approaches be developed to discover the roles of lncRNAs in the initiation and progression of RHO GTPase-mutant cancers? (iii) How can we incorporate genomic and transcriptomic data from cancer patients with RHO GTPase mutations to establish a lncRNA discovery pipeline to progress preclinical studies to clinical practice? (iv) How can we confirm that the tissue-specific expression of lncRNAs represent therapeutic candidates for tissues with a higher rate of RHO GTPase mutations?

Furthermore, various studies should be carried out to increase our understanding of the intricacies of RHO GTPase regulation with regard to lncRNAs that would be considered therapeutic targets: (i) A small number of RHO family small GTPases, such as RHOA, RAC1, and CDC42, have been well studied thus far, but there is a crucial demand to investigate the functions of other less-characterized members as well as their mechanisms of regulation. (ii) Apart from discovering more lncRNAs with sponging effects on miRNAs targeting RHO GTPases, a comprehensive study should be performed on the regulatory lncRNAs of RHO GTPase regulators, such as GDIs, GEFs, and GAPs (Figure 1). (iii) In addition to the investigation of the mechanisms of regulation of lncRNAs, the oncogenic or tumor-suppressing effects of proteins/peptides encoded by these noncoding RNAs should be clarified. (iv) The ability to target RHO GTPase-related oncogenic lncRNAs via diverse strategies, including gene-editing methods—e.g., CRISPR–Cas9 technology, nucleic acid-based drugs, catalytic degradation by ribozymes or DNAzymes, small molecule inhibitors, synthetic lncRNA and miRNA mimics, and targeting the secondary and tertiary structures of lncRNAs—should be investigated. (v) To find new drugs for abnormal RHO GTPase activity, the possible interactions of lncRNAs with their associated proteins, which could form ribonucleoprotein complexes, should be studied.

lncRNAs as the pharmaceutical targets provide a plethora of opportunities in the future. In fact, therapeutic approaches which are on the basis of the gene therapy to treat disease by artificially regulating gene expression are called “third generation” of clinical drugs. This kind of treatment is promising for controlling diseases at the genetic level, and is able to overcome the drawbacks of biological incompatible proteins. From the outlook of the clinical breakthrough in targeted mRNA drugs and the progressive recruitment for clinical trials of miRNAs, it is now conceivable that targeting lncRNAs will probably play a critical role in gene therapy in the near future. Although the significant role of the lncRNAs in the tumorigenesis and its potential clinical application in future is bright, some concerns and questions remain. The fundamental researches on the function of druggable lncRNAs and the potential downstream responses are insufficient currently; therefore, unexpected outcomes and inappropriate pathological effects may occur when lncRNA-targeted drugs are used clinically. Moreover, highly specific targeting methods and delivery systems development are required to guarantee that only the selected lncRNA is affected. Due to the insufficient data on clinical trials, the efficacy and safety of lncRNA drugs in humans remain undetermined. Therefore, further advances in lncRNA-targeted drugs are distinctly dependent on the in-depth basic research into the function and mechanisms of lncRNAs. In this regard, the diverse mode of the action repertoire of lncRNAs reveals various opportunities for their targeting. Importantly, there has been an identification of plenty of modified residues in lncRNA transcripts, indicating the epigenetic regulation of lncRNAs and its potential effects on cancer progression. For instance, m6A-modified transcripts of GAS5, as one of the regulators of RHOB, induced phosphorylation and subsequently ubiquitin-mediated degradation of YAP, leading to the inhibition of malignant transformation [276]. Exploitation of high-throughput methods—such as chromatin RNA in situ reverse transcription sequencing (CRIST-seq) for identification of lncRNAs within the regulatory elements of genes, and RNA immunoprecipitation, crosslinking, and immunoprecipitation (CLIP), as well as RNA pull-down techniques for discovering the interaction between proteomes and lncRNAs—will uncover more important roles and new mechanistic insights. An interesting progression is the detection of natural antisense transcripts (NATs): lncRNAs that are transcribed in the antisense direction to coding genes, and negatively regulate them in cis. ASOs that target NATs, called ‘antagoNATs’, have indicated very encouraging preclinical results for gene reactivation in the central nervous system. Such brilliant developments confirmed that the establishment of lncRNA-based therapeutics into clinical testing is imminent. In the case of the RHO GTPase-related lncRNAs, results showed that the subcutaneous delivery of MALAT1 phosphorothioate-modified ASO successfully suppressed primary tumor differentiation. From the perspective of the clinical benefits in RHO GTPase-related anti-lncRNA plant-derived natural compounds, credible anti-cancer activity of curcumin was reported through its downregulatory effect on H19. Tissue specific expression of several lncRNAs have also created remarkable treatment opportunities, including therapeutic manipulation of lncRNA promoters. Administration of a plasmid (BC-819) carrying the gene for diphtheria toxin under the regulation of the H19 gene promoter was utilized to investigate its anti-tumor responses in various solid tumors. A phase I/II clinical trial of intravesical BC-819 in patients with invasive bladder cancer has revealed mild and local toxicity along with complete and partial response rates of 22% and 44%, respectively. Therefore, the highly selective expression of various lncRNAs prompts the assessment of using tissue or cell-type specific lncRNA promoters to induce cytotoxic effects in disease-related cells.

Finally, a significant challenge that has not been well explored is the illumination of the gap in knowledge of spatiotemporal regulation of signal transduction and RHO GTPase activity to specifically explain the interaction of RHO GTPases with their distinct effectors and regulators. Among these regulatory mechanisms, the roles of accessory proteins and noncoding RNAs, particularly lncRNAs, in the spatiotemporal regulation of cell signaling should be fully investigated. The clarification of these important issues not only leads to the explanation of many aspects of RHO GTPase activity and their signaling pathways, but also prompts the discovery and development of novel drugs against various diseases, specifically cancer.

## Figures and Tables

**Figure 1 cancers-13-05386-f001:**
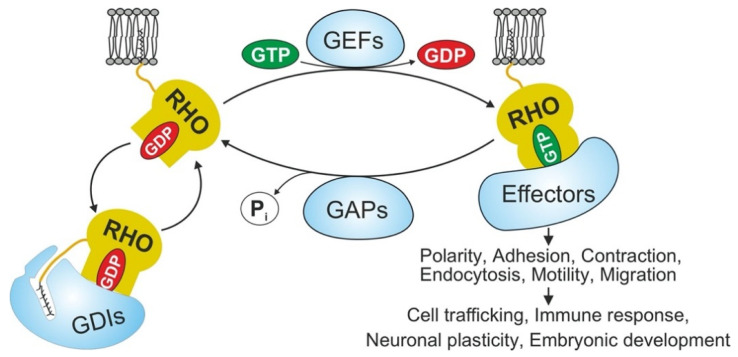
Molecular mechanisms of RHO GTPase regulation and signaling. Most of RHO GTPases (20 canonical members) act as molecular switches, in order to transduce signals from receptors to downstream pathways. This switch mechanism is tightly regulated through three classes of particular proteins: Guanine nucleotide exchange factors (GEFs: 74 DBL and 11 DOCK family members) catalyze the exchange of GDP for GTP, thereby turning on the signal transduction. GTPase-activating proteins (GAPs; 66 ARHGAP family members) bind to GTP-bound RHO GTPases and accelerate their very slow intrinsic GTP hydrolysis, leading to signaling the switch off. Guanine nucleotide dissociation inhibitor (GDIs: three known members) bind to isoprenylated, GDP-bound RHO GTPases and displace them from the membrane. GTP-bound, active RHO GTPases associate with and activate different downstream effectors (more than 70 known members), and thus accomplish various biochemical processes and cellular functions.

**Figure 2 cancers-13-05386-f002:**
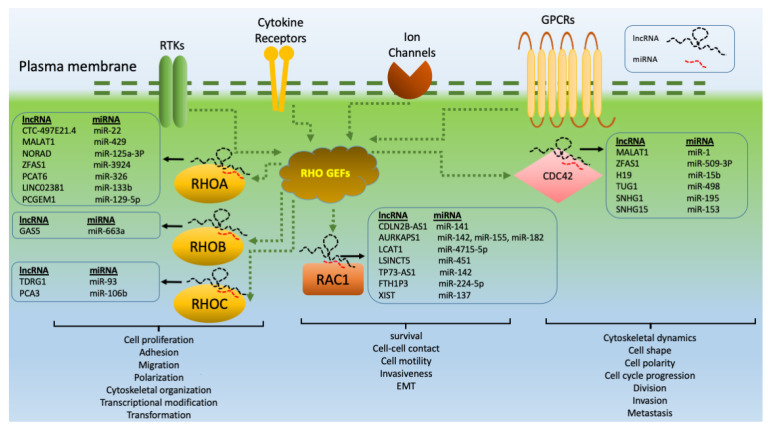
RHO GTPase-related lncRNAs. RHO GTPases are activated by RHO GEFs via different receptors, including GPCRs (G-protein coupled receptors with various ligands, such as Angiotensin II, Endothelin, LPA, S1P1, Thrombin, and Thromboxane A2), RTKs (receptor tyrosine kinases with ligands, such as EGF, Ephrins, NGF, PDGF, and VEGF), ion channels (with ligands, including nicotinic acetylcholine or glutamate), and cytokine receptors (with ligands, including Interferons, Interleukins, and Tumor necrosis factors). Members of the RHO GTPase family, such as RHOA, RHOB, RHOC, RAC1, and CDC42, are modulated by lncRNAs, which are the key regulators of gene expression. They play essential roles in a wide variety of signaling pathways, and thus processes involved in tumor progression. lncRNAs act as ceRNAs by sponging tumor suppressor miRNAs, thereby indirectly regulating the expression of genes related to RHO GTPases.

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
