# Peer review of "RHO GTPase-Related Long Noncoding RNAs in Human Cancers"

_cancers, 2021, doi:10.3390/cancers13215386_

Round 1
Reviewer 1 Report
RHO GTPase-Related Long Noncoding RNAs in Human Cancers
Mahsa Saliani, Amin Mirzaiebadizi and Mohammad Reza Ahmadian- September 2021
In this review, the authors explore the roles of Long Noncoding RNAs in regulating Rho GTPases -mediated signaling and how this regulation can contribute to tumor progression in various type of cancers. This is a widespread and extensive review, and we believe it will be of great interest to the Cancers readers once major and minor concerns are addressed. Briefly, our main concern is that while each section provides ample example of Long Noncoding RNAs-mediated regulation of individual Rho GTPase (RHOA, RHOB, RHOC, RAC1 and CDC42), it is lacking conceptual summaries on the molecular mechanism at play. We also listed minor points that prevent publication of this manuscript in its current state.
Major comment :
- The authors need to provide/state a rationale for why they decided to focus only on RHOA, RHOB, RHOC, RAC1 and CDC42. Are they the only one known to be regulated by long Noncoding RNAs to this date? Do they limit themselves to these Rho GTPases to simplify the message of the review? Whichever the reason, it needs to be clarified for the readers. One note is made in the conclusion but this need to be stated in the introduction.
- When stating a study, the authors need to clarify if the mechanism suggested is via the direct regulation of the Rho GTPase or if it is indirect. While the authors clearly stated that they aim to focus on long Noncoding RNAs as ceRNAs, it was not always clear for us if they were referring to a miRNA that is known to target the Rho GTPase directly. For example, line 180 describe a study in which the RhoGEF, NET1 is the direct target of the regulation, which indirectly impact Rho-mediated signaling.
- Each section provides an extensive list of how individual Rho GTPase are regulated by Long Noncoding RNA, yet it does not provide an integrated view or conceptual summaries of these mechanisms. To alleviate this, we suggest the authors increase/create new summary figures to interpret how an individual Rho GTPase is regulated by long noncoding RNA (for example long Noncoding RNA can regulate RHOA can be regulated via miRNA-mediated regulation of its own mRNA or via indirect regulation such as regulation of the RhoGEF, or other signaling pathway such as ROCK, which altogether impacts RHOA activity etc…
- The authors should address in their conclusion current challenges in targeting long Noncoding RNA with regards to the specificity of their targets.
Minor comment :
- Line 24, do the authors mean homeostasis?
- In line with new studies such as Golding, Bement, elife 2019, Rho GDI have also been shown to extract active Rho GTPase, therefore the authors should nuance their statement.
- Line 67 to 74, Rho GTPase are usually subdivided into 8 subfamilies, with RHOU and RHOV forming an atypical group. Also please refer to the official gene/protein name.
- Line 89, the statement regarding the recruitment of pathways to avoid antitumoral immune responses is lacking a reference.
- Paragraph starting at line 92, considering the new driver mutation identified in RAC in melanoma, which produce fast cycling Rho GTPase, the authors need to modulate their statement.
- Paragraph 128 a 135 is lacking references
- Paragraph starting at line 148 is lacking references for the various statements
- Line 176, does “by” need to be in italic?
Author Response
Manuscript ID: cancers-1393223
Point-by-point responses to the valuable comments and the constructive criticism of Reviewer #1:
In this review, the authors explore the roles of Long Noncoding RNAs in regulating Rho GTPases -mediated signaling and how this regulation can contribute to tumor progression in various type of cancers. This is a widespread and extensive review, and we believe it will be of great interest to the Cancers readers once major and minor concerns are addressed. Briefly, our main concern is that while each section provides ample example of Long Noncoding RNAs-mediated regulation of individual Rho GTPase (RHOA, RHOB, RHOC, RAC1 and CDC42), it is lacking conceptual summaries on the molecular mechanism at play. We also listed minor points that prevent publication of this manuscript in its current state.
Major comment:
1) The authors need to provide/state a rationale for why they decided to focus only on RHOA, RHOB, RHOC, RAC1 and CDC42. Are they the only one known to be regulated by long Noncoding RNAs to this date? Do they limit themselves to these Rho GTPases to simplify the message of the review? Whichever the reason, it needs to be clarified for the readers. One note is made in the conclusion but this need to be stated in the introduction.
Authors Response: Thanks for pointing to this important issue. The answer is yes, RHOA, RHOB, RHOC, RAC1 and CDC42 are the only members of the RHO family known to be regulated by long noncoding RNAs to date. To clarify for the readers why we choose RHOA, RHOB, RHOC, RAC1, and CDC42 in the context of this review article, the introduction was modified as follows: “Up to now, a number of lncRNAs related to a few RHO GTPase members, such as RHOA, RHOB, RHOC, RAC1, and CDC42, have been identified.”
2) When stating a study, the authors need to clarify if the mechanism suggested is via the direct regulation of the Rho GTPase or if it is indirect. While the authors clearly stated that they aim to focus on long Noncoding RNAs as ceRNAs, it was not always clear for us if they were referring to a miRNA that is known to target the Rho GTPase directly. For example, line 180 describe a study in which the RhoGEF, NET1 is the direct target of the regulation, which indirectly impact Rho-mediated signaling.
Authors Response: We greatly appreciate this suggestion. We now added a new explanation in the context to demonstrate that all mentioned lncRNAs in this article act as ceRNAs in the direct regulation of specific Rho GTPases. In addition, we added the following new paragraph (Title: Indirect regulatory effects of lncRNAs on RHO GTPases) on page 13, line 603, which also describes the indirect regulatory effect of ceRNAs on RHO GTPases, and transferred ceRNAs with indirect effect, such as CTC-497E21.4 to this new part.
3) Each section provides an extensive list of how individual Rho GTPase are regulated by Long Noncoding RNA, yet it does not provide an integrated view or conceptual summaries of these mechanisms. To alleviate this, we suggest the authors increase/create new summary figures to interpret how an individual Rho GTPase is regulated by long noncoding RNA (for example long Noncoding RNA can regulate RHOA can be regulated via miRNA-mediated regulation of its own mRNA or via indirect regulation such as regulation of the RhoGEF, or other signaling pathway such as ROCK, which altogether impacts RHOA activity etc…
Authors Response: Thanks a lot for raising this central point. The former Figure 1 (now Figure 2) basically illustrates the sponging effects of the RHO GTPase-specific lncRNAs that sequester the respective target miRNAs and thus basically maintain RHO GTPase signaling. By contrast, indirect regulatory effects of lncRNAs on RHO GTPase signaling is explained in new paragraph on page 13, line 624.
4) The authors should address in their conclusion current challenges in targeting long Noncoding RNA with regards to the specificity of their targets.
Authors Response: Thanks for pointing to this interesting issue. We have included a new paragraph in this regard dealing with “lncRNAs as novel therapeutic targets” (page 14, line 684).
Minor comment:
1) Line 24, do the authors mean homeostasis?
Authors Response: We changed “cell hemostasis” to “cell homeostasis” on page 1, line 24.
2) In line with new studies such as Golding, Bement, elife 2019, Rho GDI have also been shown to extract active Rho GTPase, therefore the authors should nuance their statement.
Authors Response: We added “RhoGDI can extract both inactive and active RhoGTPases, and found that extraction of active RhoGTPase contributes to their spatial regulation around cell wounds (11).” on page 2, line 56.
3) Line 67 to 74, Rho GTPase are usually subdivided into 8 subfamilies, with RHOU and RHOV forming an atypical group. Also please refer to the official gene/protein name.
Authors Response: We changed “RHO (RHOA, RHOB, and RHOC); RAC (RAC1, RAC1B, RAC2, RAC3, and RHOG); CDC42 (CDC42, G25K, TC10, TCL, WRCH1, and WRCH2); RHOD (RHOD and RIF); RND (RND1, RND2, and RND3); and RHOH” to “RHO (RHOA, RHOB, and RHOC); RAC (RAC1, RAC1B, RAC2, RAC3, and RHOG); CDC42 (CDC42, G25K, TC10/RHOQ, TCL/RHOJ, WRCH1/RHOU, and WRCH2/RHOV); RHOD (RHOD and RIF/RHOF); RND (RND1/RHO6, RND2/RHO7, and RND3/RHO8/RHOE); and TTF/RHOH” on page 2, line 87 by additionally referring to official protein names.
4) Line 89, the statement regarding the recruitment of pathways to avoid antitumoral immune responses is lacking a reference.
Authors Response: The reference [33] is added on page 3, line 106.
5) Paragraph starting at line 92, considering the new driver mutation identified in RAC in melanoma, which produce fast cycling Rho GTPase, the authors need to modulate their statement.
Authors Response: This part is extended with “Contrary to recent studies that report the identification of new driver mutations in some RHO GTPases members, such as RAC1, RHOA, and CDC42 [34].” by referring to the identification of the new driver mutations on page 3, lines 110 and 111.
6) Paragraph 128 a 135 is lacking references
Authors Response: New references [62] and [63] and [64] are added on page 4, lines 149, 150 and 151.
7) Paragraph starting at line 148 is lacking references for the various statements
Authors Response: A reference [73] and [74] and [76] are added on page 4, line 173 and page 5, lines 175, and 177.
8) Line 176, does “by” need to be in italic?
Authors Response: It was Corrected.
Reviewer 2 Report
In the present review article entitled ‘RHO GTPase-Related Long Noncoding RNAs in Human 2 Cancers’, authors discuss the role of lncRNAs in regulating a set of RHO GTPases in cancer and the associated signaling. This review suffers from few limitations. My comments are appended as below:
- In the first place, it would be vague to refer to as ‘cancer’. Authors should stick or certain cancer type and describe explicitly.
- Authors should represent a figure explaining the basic mechanism,, including upstream and downstream components of RHO GTPases signaling.
- Authors should include a table describing the RHO GTPases superfamily members along with the components involved in its regulation.
- While describing the miRNA and RHO GTPases, authors should provide a representative example and describe.
- Line 108-109- cite the particular name of lncRNA and cancer type.
- Authors should explicitly describe the study with patients, describing the number and the statistical inference.
- The authors should describe the cross-talk between RHO GTPases associated components.
- As RHO GTPases are also localized in the nucleus, does it have any role in epigenetic modifications?
- Each section should end with a summary statement.
- Authors should provide the list of inhibitors approved by the FDA or under trial in tabular form.
- There should be a ‘future directions’ section. I observe that authors do include open questions in the conclusion section, but ‘future directions’ need to be highlighted.
Author Response
Manuscript ID: cancers-1393223
Point-by-point responses to the valuable comments and the constructive criticism of Reviewer #2:
In the present review article entitled ‘RHO GTPase-Related Long Noncoding RNAs in Human 2 Cancers’, authors discuss the role of lncRNAs in regulating a set of RHO GTPases in cancer and the associated signaling. This review suffers from few limitations. My comments are appended as below:
1) In the first place, it would be vague to refer to as ‘cancer’. Authors should stick or certain cancer type and describe explicitly.
Authors Response: Thanks a lot for pointing to this critical issue. We have indeed referred to as “cancer” many times without specifying the cancer types or writing instead “tumorigenicity”. These text passages (lines 11, 108, 126, 304, 487, and 572) are now revised.
2) Authors should represent a figure explaining the basic mechanism, including upstream and downstream components of RHO GTPases signaling.
Authors Response: We greatly appreciate this suggestion. A figure (new Figure 1) is now added to revised manuscript that schematically highlights the regulators and effector signaling. explain the basic mechanisms of RHO GTPase regulation and signaling.
3) Authors should include a table describing the RHO GTPases superfamily members along with the components involved in its regulation.
Authors Response: Thanks for this suggestion, which would be important if we review lncRNAs related to regulators and effectors of RHO GTPases. However, only a few references were described in the current article, which describe “Indirect regulatory effects of lncRNAs on RHO GTPases”. Therefore, we added a new paragraph on page 13, line 624, which describes the effects of lncRNAs related to a RHOA GEF and RHOA effectors.
The RHO GTPases superfamily members are now completed page 2, line 84.
4) While describing the miRNA and RHO GTPases, authors should provide a representative example and describe.
Authors Response: Thanks for pointing to this issue. For all of the studies describing the miRNA and RHO GTPases, representative examples in the respective cancer types, for example PCA3/miR-106b-5p/RHOC axis in epithelial ovarian cancer, are added and described.
5) Line 108-109- cite the particular name of lncRNA and cancer type.
Authors Response: Lines 108-109 are not about lncRNAs. So, we think that comment is related to lines 188 and 189. In this regard, studies on the particular cancer types and the respective lncRNAs are now cited on page 5, line 188-189.
6) Authors should explicitly describe the study with patients, describing the number and the statistical inference.
Authors Response: Thanks for pointing to this issue. All of the studies with patients were now described in more details (For example, see pages 6, 7, and 8).
7) The authors should describe the cross-talk between RHO GTPases associated components.
Authors Response: We are thankful for this recommendation. Cross-talk between RHO GTPases can take place at different stages, and involve certain protein complexes. A great deal of our knowledge stems from the studies of localization and activities of RHO GTPase-specific GEFs and GAPs (Nimnual et al., 2003; Zenke et al., 2004). We have referred to these and our previous studies, which have described RHO GTPases associated components in detail (Jaiswal et al., JBC, 2011; Jaiswal et al., JBC, 2013; Jaiswal et al., JBC, 2014; Amin et al., JBC, 2016; Mosaddeghzadeh et al., Cells, 2021). The new Figure 1 in the manuscript summarizes the regulatory components of RHO GTPases as well as the downstream effectors.
8) As RHO GTPases are also localized in the nucleus, does it have any role in epigenetic modifications?
Authors Response: We thank the reviewer for this point. Indeed, the activity of some RHO GTPases is associated with their nuclear localization, including the role of (1) nuclear RAC1 in nuclear plasticity and tumor cell invasion (Disanza and Scita, Dev cell, 2015; Navarro-Lerida et al, Dev cell, 2015), (2) nuclear RHOA and its specific GEF NET1 in DNA damage response (Dubash et al, PLOS One, 2011), and (3) nuclear CDC42 and its GEF ECT2 in the stabilization of CENP-A at centromeres in breast cancer cells (Zhang et al., Theranostics, 2020). To our knowledge, there is no direct connection of these nuclear proteins in epigenetic modification but lncRNAs do so. A new paragraph explaining the role of the lncRNAs in epigenetic modifications is now included into the revised manuscript (page 14) with some examples about the potential role of the lncRNAs in epigenetic regulation of RHO GTPases.
9) Each section should end with a summary statement.
Authors Response: Thanks for pointing to this issue. A summary statement was added to each section.
10) Authors should provide the list of inhibitors approved by the FDA or under trial in tabular form.
Authors Response: We could not find studies on FDA-approved compounds targeting lncRNAs. Wang et al. have reported that lncRNAs play important roles in cancer drug response by investigating the effects of 49 FDA approved drugs to the 5605 tumor samples from 21 cancer types (Wang et al., Nat Commun, 2018). However, we have thought of including a new paragraph in this regard dealing with “lncRNAs as novel therapeutic targets” (page 14).
11) There should be a ‘future directions’ section. I observe that authors do include open questions in the conclusion section, but ‘future directions’ need to be highlighted.
Authors Response: Thanks a lot for this clear point. We had several aspects regarding the future directions, which are now emphasized with additional sections at the end of the revised manuscript (on pages 13 and 14). To be more accurate, we changed the subheading “conclusion” to “Future Perspectives”.
Round 2
Reviewer 2 Report
I congratulate the authors for providing the modifications. With this, the manuscript is closer to publication. I, however, request to answer the following minor points:
- Figure 1- I suggest authors add few descriptive details as to which pathways and processes are dominantly altered? Please specify the key biological functions.
- Figure 2- please depict the ligands.
- Question 6- please explicitly mention the number of patients and add the statistical inference (HR, P-value as available)
- In future directions, as authors mention lncRNAs pharmacological targets, I wonder if authors justify with candidate examples.
Author Response
[Cancers] Manuscript ID: cancers-1393223
Authors Responses to the comments of Reviewer #2:
Thank you for your time, effort, and valuable criticism, which has considerably improved the quality and readability of our manuscript.
Hereby, we have responded to your comments point-by-point:
- Figure 1- I suggest authors add few descriptive details as to which pathways and processes are dominantly altered? Please specify the key biological functions.
Authors‘ response: We have included more details considering the pathways and biological functions controlled by RHO GTPases in Figure 1.
- Figure 2- please depict the ligands.
Authors‘ response: Information about receptors’ ligands were added in to the legend of Figure 2.
- Question 6- please explicitly mention the number of patients and add the statistical inference (HR, P-value as available)
Authors‘ response: The details about the number of the patients and statistical inference were added into the manuscript.
- In future directions, as authors mention lncRNAs pharmacological targets, I wonder if authors justify with candidate examples.
Authors‘ response: Examples regarding the pharmacological targeting of RHO GTPase-related lncRNAs, including GAS5, MALAT1, and H19 were added to the section future perspectives.